# Manual Abilities and Cognition in Children with Cerebral Palsy: Do Fine Motor Skills Impact Cognition as Measured by the Bayley Scales of Infant Development?

**DOI:** 10.3390/bs13070542

**Published:** 2023-06-29

**Authors:** Thais Invencao Cabral, Xueliang Pan, Tanya Tripathi, Jianing Ma, Jill C. Heathcock

**Affiliations:** 1School of Health and Rehabilitation Sciences, The Ohio State University, Columbus, OH 43220, USA; thais.invencaocabral@osumc.edu (T.I.C.); tanya.tripathi@mcri.edu.au (T.T.); 2Department of Biomedical Informatics, The Ohio State University, Columbus, OH 43210, USA; jeff.pan@osumc.edu (X.P.); jianing.ma@osumc.edu (J.M.)

**Keywords:** child, cognitive impairment, psychomotor performance, disability, upper extremity, manual ability

## Abstract

Manual ability may be an important consideration when measuring cognition in children with CP because many items on cognitive tests require fine motor skills. This study investigated the association of fine motor dependent (FMD) and fine motor independent (FMI) items within the cognitive domain (COG) of the Bayley Scales of Infant Development—Third Edition (Bayley-III) and Manual Ability Classification System (MACS) in children with cerebral palsy. Children aged 2 to 8 (3.96 ± 1.68) years were included in this study. MACS levels were assigned at baseline. COG was administrated at baseline (n = 61) and nine months post-baseline (n = 28). The 91 items were classified into FMD (52) and FMI (39). Total raw score, FMD, and FMI scores were calculated. The association between MACS and cognitive scores (total, FMD, and FMI) were evaluated using linear regression and Spearman correlation coefficients. We found total, FMD, and FMI scores decrease significantly as the MACS level increases at the baseline. Both FMD and FMI scores decreased as MACS levels increased (worse function). There was a significant difference between the two slopes, with the FMD scores having a steeper slope. Similar patterns were observed nine months post-baseline. Children with lower manual ability scored lower in the cognitive domain at baseline and 9 months post-baseline. The significant difference in the performance of FMD items and FMI items across MACS levels with a steeper slope of changes in FMD items suggests fine motor skills impact cognition.

## 1. Introduction

Cerebral palsy (CP) is a heterogeneous group of neurodevelopmental conditions that present as impairments in movement and posture. CP is the result of brain injury during fetal or infant development. Brain injuries that cause motor impairments are permanent and non-progressive. It is often accompanied by epilepsy, secondary musculoskeletal disorders, and sensory and cognitive impairments, which cause limitations in activities and participation [1,2,3,4,5]. CP is commonly classified according to muscle tone topography [6], gross motor function [7], and manual function [8].

Functional impairments across developmental domains in children with CP are often evaluated using standardized assessments of cognitive and motor skills [9,10,11,12]. Many standardized cognitive assessments appropriate for children with cerebral palsy, such as the Bayley Scales of Infant Development—Third Edition (Bayley-III) and the Weschler Intelligence Scale for Children, require fine motor actions to accomplish cognitive tasks. Most cognitive assessments require reaching, pointing, grasping, manipulating objects, and other manual abilities that demand precision, speed, dexterity, and coordination [13,14,15,16,17]. Thus, whether the cognitive scores reflect purely cognitive performance or if cognitive performance is masked by poor fine motor skills is sometimes unclear, especially in children with limited fine motor or manual skills [17].

In children with CP, cognitive impairment is present in 50% of the cases [2,18,19]. Cognitive impairment is influenced by many components, such as the type and distribution of CP, gross motor functioning, and manual ability [18,19]. Bilateral spastic CP is common and represents 60 to 80% of the occurrences, with 40.3% of the children classified at GMFCS levels IV and V. Upper extremity functions are impacted in a range from 57 to 83% of the occurrences [2]. Additionally, impairments in manual function may cause disuse and lack of learning opportunities early in life, i.e., impairments in upper extremity function limit opportunities for manipulation and object exploration, which have an impact on cognitive development [17,20]. Also, motor and cognitive development overlap and are interrelated [21]. According to Osorio-Valencia et al. [14], the components of cognitive skill are influenced by gross and fine motor abilities acquired in the first three years of life. However, the long-term interrelation of fine motor and cognitive skills still needs to be clarified, especially for children with motor disabilities.

Children with CP might be unable to show their abilities compared to neurotypical children in the cognitive domain during the standardized assessment [20,22,23,24] because of the requirements in fine motor skills. They may need more time to deal with the material or might not be able to manipulate the test materials without adaptation. The study by Visser et al. [16,17] examined the validity of the Bayley-III Low Motor/Vision version and its suitability for children with motor and/or visual impairment(s). It contains accommodated items, that is, adaptations to minimize impairment bias, without altering what the test measures. The results found that the accommodations in the cognitive domain did not affect the test scores of children with neurotypical development and did improve the test scores of children with atypical development. In addition, the results indicated that most children with atypical development could show their abilities in the cognitive domain and that the accommodations were beneficial in 29 of these 52 cases. Therefore, standardized tools to evaluate cognition that consider adaptations, especially for children with manual ability impairment are important [16,17,22,23,24].

To evaluate cognition in children with CP, understanding the relationship between manual and cognitive abilities is important, especially when administering a standardized assessment such as Bayley-III. Measuring cognition in children with CP while considering their manual ability and the demands of the test could provide a more accurate method for tracking development, predicting outcomes, and evaluating intervention approaches to provide accurate and reliable quantifications of performance. Besides that, an accurate measure and reporting can facilitate a comprehensive discussion between parents and rehabilitation professionals on the children’s cognitive abilities.

The primary purpose of this study was to identify the association between manual ability and cognition in children with CP. Our goal was to investigate if manual ability levels and cognitive performance in children with CP were related to fine motor-dependent and -independent items within the cognitive domain of Bayley-III. These categories of fine motor dependent and independent were assigned by our research team. The secondary purpose was to investigate the impact of the distribution of CP and gross motor function levels (GMFCS) and factors such as gestational age, birth weight, NICU stay, age of CP diagnosis, and CP type on the relationship between manual ability levels and cognition. We expected children with lower manual ability to have lower cognitive scores than children with higher manual ability. Additionally, we anticipated that gross motor functions would be associated with manual and cognitive abilities.

## 2. Materials and Methods

### 2.1. Participants

Participants in this prospective, observational study were children with CP who were part of a larger randomized controlled clinical trial [NCT02897024]. Sixty-one children aged between 2 and 8 years (median age: 3.50; IQR: 2.58, 4.75) participated in this study. The Ohio State University and Nationwide Children’s Hospital’s [NCH] Institutional Review Boards approved this study [IRB16-00492], and parental consent was obtained for all participants. Participants with uncontrollable seizures, unknown auditory or visual impairments, progressive neurological disorder, recent surgery, or participation in another daily physical therapy treatment program in the last six months were excluded from the larger trial, and only those eligible for the Bayley evaluation were included in this study. See Table 1 for demographic information.

### 2.2. Procedure

The participants were assessed at baseline and nine months post-baseline. At baseline, the manual ability level of children with CP was classified using Mini-Manual Ability Classification System (mini-MACS) or MACS according to the participant’s age. The MACS describes children with CP 4 to 18 years of age in daily manual activities. Mini-MACS is an updated version of the MACS for younger children 1 to 4 years of age but has the same concept as MACS. In this study, the term MACS will be used for both classification systems. The cognitive domain of the Bayley Scales of Infant Development—Third Edition (Bayley III) was administered at baseline (n = 61) and nine months post-baseline (n = 28).

The participants received 40 h of outpatient physical therapy during the nine months with a randomized treatment service delivery frequency: daily (high intensity periodic) or weekly (usual and customary treatment). No between-group treatment effects were expected or found between groups for cognition.

### 2.3. Measures

#### 2.3.1. Manual Ability Classification System (MACS)

The Manual Ability Classification System (MACS) describes how children with CP 4 to 18 years of age use both hands together to manipulate toys and objects in daily activities. MACS is described hierarchically in five levels (I to V). The levels are based on the self-initiated ability to handle objects and the need for assistance or adaptation to perform manual activities [8]. Children in level I (highest ability) can easily and successfully handle objects. In contrast, children in level V (most limited ability) cannot handle objects or complete simple manual actions alone [8]. We used the Mini-MACS as the manual classification system for those under four.

MACS is a classification system, not an outcome measure. It was used to classify our sample of participants. Trained, reliable, and blinded assessors determined the MACS level for the participants. The intra- and inter-rater correlation coefficients were calculated for each blinded assessor every six months. All assessors needed to achieve and maintain the agreement index >85% to pass reliability. The agreement index was evaluated using inter-rater reliability (IRR) in 10% of the sample.

#### 2.3.2. Bayley Scales of Infant and Toddler Development—Third Edition (Bayley-III)

The Bayley-III is a valid and reliable measure of a child’s neurodevelopment from 1 to 42 months of age and was developed specifically for use in research and clinical practice. It is the most common assessment tool for evaluating early development and measuring delays across multiple domains of development (cognition, motor, language, and socio-emotional) [12]. Bayley-III has been validated for use in children at high risk for CP with good discriminative properties [12,23,24]. According to the manual administration’s instructions, Bayley-III was administered to children out of the age range but in the developmental range appropriate for this tool [12]. The raw score was considered in the analysis. Blinded assessors completed training and intra- and inter-rater reliability testing on the Bayley-III (>85% to pass reliability).

The cognitive total raw score was based on 91 cognitive domain items. These 91 items were classified into two groups (see Appendix A
Table A1): 52 items relied on fine motor abilities (reaching, pointing, grasping, and manipulating objects, with components such as precision, speed, dexterity, and coordination), which were classified as fine motor dependent (FMD); 39 items not requiring the skills listed (looking at pictures, turning the head to specific sounds, counting numbers) were classified as fine motor independent (FMI). The items were classified by two experienced researchers trained in Bayley-III, with a strong agreement between the researchers. The FMD and FMI scores were calculated as the proportion of the number of items scored, specifically, the sum of the items that received credit (score of 1) divided by the total number of items (52 and 39 for FMD and FMI, respectively). In other words, the FMD score is the total FMD items credited/52, and the FMI score is the total FMI items credited/39.

### 2.4. Analysis

Descriptive statistics (means and standard deviation) of the Bayley-III cognitive raw scores at baseline and nine months post-baseline were summarized for children at each MACS level. The Spearman correlation coefficient was used to evaluate the association between MACS levels and cognitive scores at baseline and nine months post-baseline. General Linear Model and Kruskal–Wallis tests were used to explore the difference in the cognitive scores (including total raw score and proportion of fine motor dependent and independent scores) among MACS levels at baseline. The change in the cognitive score from baseline to nine months post-baseline and the impact of the baseline MACS levels were explored using linear mixed models to account for the association of the baseline and nine months post-baseline data from the same participant. All 61 participants were included in these analyses while assuming data missing at random for these without post-baseline data.

In addition, using multiple regression analysis, we further investigated the impact of the distribution of CP and gross motor function level (Gross Motor Function Classification—GMFCS) and factors such as gestational age, birth weight, NICU stay, age of CP diagnosis, and CP type on the association between MACS level and Bayley-III cognitive scores (total scores, FMD, FMI) at baseline. See Table 2 for descriptive statistics (means and standard deviation) of the Bayley-III cognitive raw scores at baseline and nine months post-baseline.

## 3. Results

There was a significant association between the MACS levels and the Bayley–III cognitive total raw scores of children with CP at baseline (Spearman correlation Rho = −0.84 and *p* values of <0.001). The higher the MACS level (lower manual ability), the lower the cognitive scores. See Figure 1 for Bayley-III cognitive total scores at baseline and 9-month post-baseline across the MACS levels. A similar relationship between cognitive score and MACS levels was observed nine months post-baseline (Rho = −0.80 and *p* values of <0.001).

At baseline, cognitive scores of children with CP in both the FMD and FMI groups were significantly associated with their MACS levels (Rho_FMD_ = −0.85, *p* < 0.001 and Rho_FMI_ = −0.81, *p* < 0.0001). See Figure 2 for the FMD and FMI scores across the MACS levels. The decline between FMD and FMI cognitive scores over MACS levels was significantly different (*p* < 0.001). Additionally, the decline of FMD scores across MACS levels is steeper than the FMI scores (0.09 vs. 0.187 for FMD and FMI, respectively, linear regression models, Figure 2). Similar patterns in the change slope in FMD and FMI cognitive scores over MACS levels were observed nine months post-baseline.

Medical and demographic factors impact manual abilities and cognition in children with CP at baseline.

A significant association was found between the CP distribution and MACS level (*p* < 0.001). Moreover, the CP distribution was significantly related to the cognitive total raw score, FMD score, and FMI score (*p* < 0.001, *p* < 0.001, and *p* = 0.002, respectively) items. We did not find significant associations at nine months post-baseline. Similarly, GMFCS levels were also associated with MACS level and cognitive scores (total, FMD, and FMI). These findings suggest that a combination of better manual ability and gross motor function corresponds to higher cognitive performance in children with CP. In addition, the gestational age, birth weight, NICU stay, age of CP diagnosis, and CP type did not significantly impact the association between MACS level and Bayley-III cognitive scores (total scores, FMD, FMI) at baseline. See Table 3 for the Bayley-III cognitive total raw score and FMD and FMI items across the MACS level amongst the CP distribution.

## 4. Discussion

This study aimed to identify the association between manual ability and cognitive performance in children with CP. In addition, the aim was to investigate if manual ability levels and cognitive performance of children with CP were related to cognitive items on the Bayley-III that require fine motor components (fine motor dependent) and items that do not require fine motor components (fine motor independent) within the cognitive domain items of the Bayley-III. The second analysis investigated the impact of the CP distribution and gross motor function levels (GMFCS) and factors such as gestational age, birth weight, NICU stay, age of CP diagnosis, and CP type on the relationship between manual ability levels and cognitive performance. Our hypotheses were partially supported.

Manual exploration affords learning opportunities that impact cognition in children with CP by manipulating objects [10,11]. Our findings show an association between manual ability and cognition in this population. Specifically, lower manual ability (higher MACS level) corresponds to lower cognitive performance. Due to the manual ability impairment (for example, not being able to grasp an object), children with CP might demonstrate difficulty in cognitive tasks, such as solving a puzzle, within the cognitive domain of Bayley-III. Our findings support the interrelationship between movement and learning [10,11,15,19,25,26,27,28,29,30], which are basic principles for building cognitive functions [18,31,32]. Our findings emphasized that the cognitive abilities of children with CP are likely underestimated in Bayley-III due to the inherent reliance on the hands and arms to complete cognitive tasks. This underestimation was particularly evident for children with severe manual impairment, as indicated by MACS levels IV and V. Returning to our initial question of whether the cognitive scores reflect purely cognitive performance or are influenced by manual ability, our findings suggest that manual ability has an impact and can mask cognitive performance in children with CP.

The manual ability level and the success in performing cognitive tasks that have or do not have fine motor components are related according to our findings. Children with CP in MACS levels IV and V had significantly lower scores in the cognitive items, dependent and independent of fine motor abilities, than those in MACS levels II and I. However, our findings suggest a large difference between FMD and FMI across MACS levels III to V, as observed in Figure 2, suggesting their cognitive performances are lower on FMD items than on FMI items. Thus, the manual ability has a higher impact on the fine motor-dependent items than on the fine motor-independent items, especially for those with severe manual impairments. In this study, children in MACS levels I and II performed better than children in MACS IV to V because in Bayley-III, children with MACS levels I to III commonly use their less-affected hand for the cognitive items and can, presumably, score within the normative reference values, regardless of the impairment with the affected hand. However, children in MACS levels IV and V, with both upper extremities impaired, will likely perform lower and consequently score lower in the cognitive domain. Although children with very low manual abilities struggled to perform well in cognitive items requiring fine motor abilities, they also struggled in cognitive items not requiring them. This finding could be attributed to the fact that child development is a product of multi-domain interactions [18,19,25,27,30,31,32], and thus, limitations beyond manual abilities across development impact cognitive performance. In addition, cognitive impairments are common in this population.

The findings of this study also demonstrate that cognitive performance in children with CP changes over time. In this study, despite the challenges in manual functions, children with CP at different levels of manual ability improved their cognitive performance. Considering the CP distribution factor, our findings suggest that children with hemiplegia and diplegia CP, distributed mostly on MACS levels I to III, performed better on fine motor dependent items and, consequently, had better cognition performance than children with triplegia or quadriplegia, mostly assigned to levels IV and V. Children with triplegia or quadriplegia seem to perform better on fine motor independent items. Previous studies have shown that up to 29% of children with CP may be incapable of demonstrating their actual cognitive capacity in most standardized assessments, such as the commonly used Bayley-III, due to poor fine motor skills or verbal demands involved in completing most of the testing items [5,23,24,30,32]. Our study demonstrates that 43% of our sample might have their cognitive abilities misrepresented due to their manual impairment, as seen in Table 3. Thus, this study has an emerging answer to whether cognitive scores reflect purely cognitive performance or are influenced by fine motor ability, but deep investigations are needed. These data suggest that the 52 cognitive fine motor dependent items are more appropriate for children with diplegia and hemiplegia than for those with quadriplegia and triplegia.

Cognitive total raw scores were significantly different across children’s gross motor function levels. Gross motor and manual exploration support learning opportunities that further impact cognition in children with CP [10,11,25]. Our findings suggest lower cognitive performance is related to lower manual abilities (higher MACS level). Besides that, cognitive performance and manual abilities are significantly associated with gross motor function. Through the MACS and GMFCS level documentation, a snapshot of the cognitive challenges of children with CP can be anticipated. The combination of MACS and GMFCS when referring to cognitive performance is especially relevant for children with severe limitations in manual ability. This study contributes to the clinical practice field, reinforcing that a combination of classification systems, such as MACS and GMFCS, and CP distribution need to be included in the assessment and individualized treatment plans for cognition. These will provide accurate, reliable quantifications and facilitate a comprehensive discussion between parents and rehabilitation professionals on understanding real-life barriers children with CP face. In addition, raising awareness of these findings is substantial regarding the cognitive performance of children with CP, especially at MACS and GMFCS levels IV and V. Thus, when reporting cognitive performance in pediatric services, motor abilities should be accounted for interpreting cognition, especially for children with significant impairments.

To our knowledge, this is the first study to categorize the testing items in the cognitive domain of Bayley-III based on if fine motor skills are required to receive a full score and then compare the performance of children with CP who have different manual abilities (MACS level) on these item categories (fine motor dependent versus fine motor independent). This study highlights that evaluating cognitive performance in children with CP using standardized tools such as Bayley-III needs careful interpretation and modifications. For children with CP, an assessment tool for cognitive performance could consider (1) the MACS levels when interpreting the scores, (2) different attributions for FM-dependent and FM-independent items, (3) adjustment on the requirements of the tasks such as timing the items, and/or (4) bimanual and unimanual skill necessary to complete items. Previous studies, such as Visser et al. [16,17], have demonstrated success with valid results in the accommodations made on Bayley-III. The accommodations were beneficial for a subset of children with atypical development who showed a larger raw score. Thus, our study adds that accommodations might be needed for children with CP on cognitive scales, especially in MACS IV and V. Cognitive performance affects daily functioning and predicts participation and is an important factor when addressing the treatment plan. Future research is necessary to review the current instruments available to evaluate cognitive performance and develop an appropriate standardized instrument for children with atypical development as children with CP.

There are some limitations to this current study. First, this study is an exploratory analysis of a set of data from a large clinical trial and other components of the trial may have confounded the results. Second, some children in the age range at baseline were not in the developmental range nine months later, meaning that the sample size is smaller than the baseline analysis, which might have impacted our ability to assess if differences in manual abilities account for the magnitude of change in cognitive performance. Third, our sample had unequal sample sizes among MACS levels, possibly resulting in lower power for the subgroup analysis. Fourth, although a comprehensive list of factors (birth-related, medical, and environmental) was analyzed, other demographics, such as parental education and maternal age, were not considered.

## 5. Conclusions

Children with CP are vulnerable to motor and cognitive impairments [1,23,24,29]. Fine motor and cognition functions develop concurrently in children with CP, where deficits in manual abilities may also indicate cognitive struggles. Categorizing the cognitive domain items of Bayley-III that require fine motor skills and comparing the performance of children with CP who have different manual abilities on the items fine motor dependent and independent is a novel approach. Understanding the relationship between manual abilities and cognition testing items may help healthcare professionals identify children’s potential with CP. Our findings elaborate on the need for a deep investigation into whether cognitive scores reflect purely cognitive performances or are influenced by fine motor abilities.

## Figures and Tables

**Figure 1 behavsci-13-00542-f001:**
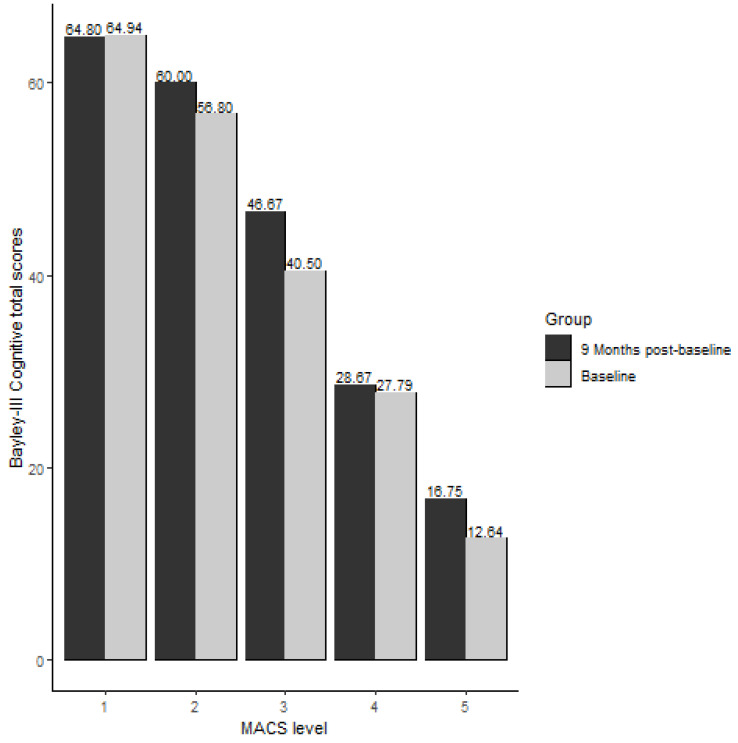
Bayley-III cognitive total scores at baseline and 9-month post-baseline via MACS level.

**Figure 2 behavsci-13-00542-f002:**
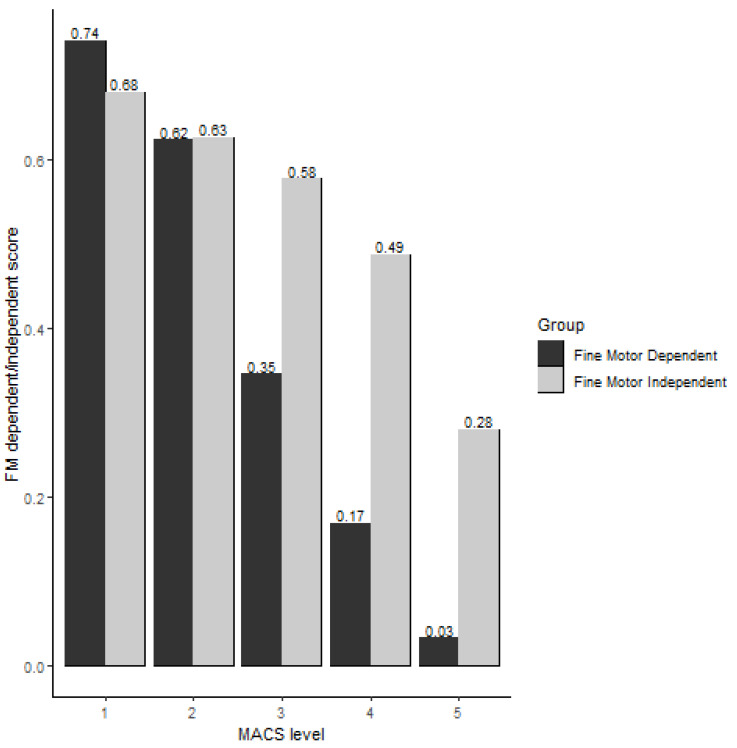
Fine motor dependent and fine motor independent score via MACS level (y-axis as FM dependent/independent score).

**Table 1 behavsci-13-00542-t001:** Baseline participant characteristics (median and interquartile for continuous variables, count and % for categorical variables).

Characteristic	Overall (n = 61)
Gestational Age (weeks)	37.0 (27.1, 39.0)
Birth Weight (kg)	2.48 (0.93, 3.41)
Birth Length (in)	18.0 (13.0, 20.0)
APGAR 1	6.00 (2.00, 8.00)
APGAR 5	6.00 (5.00, 9.00)
Total Hospital Length of Stay (days)	28.50 (0.00, 134.75)
Type of CP	
Hypotonic	10 (16.39%)
Hypertonic Spastic	45 (73.77%)
Ataxic	5 (8.20%)
Unspecified	1 (1.64%)
CP distribution	
Left hemiplegia	9 (15.52%)
Right hemiplegia	2 (3.45%)
Diplegia	9 (15.52%)
Quadriplegia	37 (63.79%)
Triplegia	1 (1.72%)
Not reported	3
GMFCS	
Level I	14 (23.33%)
Level II	8 (13.33%)
Level III	5 (8.33%)
Level IV	21 (35.00%)
Level V	12 (20.00%)
Age at enrollment (years)	3.50 (2.58, 4.75)
Gender	
Male	37 (60.66%)
Female	24 (39.34%)
Race	
White	43 (70.49%)
Black or African American	12 (19.67%)
More than One Race	4 (6.56%)
Asian	2 (3.28%)
Hispanic	1 (1.64%)

**Table 2 behavsci-13-00542-t002:** Bayley-III cognitive score at baseline and 9-month post-baseline via MACS level (mean and standard deviation).

MACS Level	Baseline (n = 61)			9-Month Post-Baseline (n = 28)		
	**Total**	**FMD Score ***	**FMI Score ****	**Total**	**FMD Score ***	**FMI Score ****
1	64.94 (14.10)	0.74 (0.20)	0.68 (0.10)	64.80 (15.28)	0.73 (0.21)	0.68 (0.11)
2	56.80 (16.40)	0.62 (0.24)	0.63 (0.11)	60.00 (12.88)	0.68 (0.17)	0.63 (0.10)
3	40.50 (14.20)	0.35 (0.23)	0.58 (0.06)	46.67 (21.36)	0.44 (0.33)	0.61 (0.10)
4	27.79 (8.14)	0.17 (0.09)	0.49 (0.10)	28.67 (10.03)	0.17 (0.14)	0.50 (0.07)
5	12.64 (8.82)	0.03 (0.07)	0.28 (0.15)	16.75 (13.30)	0.07 (0.10)	0.34 (0.23)

* FMD score = total FMD items credited/52; ** FMI score = total FMI items credited/39.

**Table 3 behavsci-13-00542-t003:** Cerebral palsy distribution and GMFCS level across MACS level at baseline for the Bayley-III cognitive total raw score and fine motor dependent and independent items.

Cerebral Palsy Distribution					GMFCS Level					
	Hemiplegia (n = 11)	Diplegia (n = 9)	Quadriplegia and Triplegia (n = 38)	*p*-Value	I	II	III	IV	V	*p*-Value
MACS Level				<0.001						NA ^1^
1	6 (55%)	6 (67%)	3 (7.9%)		11 (79%)	3 (38%)	1 (20%)	1 (4.8%)	0 (0%)	
2	4 (36%)	2 (22%)	8 (21%)		3 (21%)	4 (50%)	3 (60%)	5 (24%)	0 (0%)	
3	0 (0%)	1 (11%)	3 (7.9%)		0 (0%)	1 (12%)	0 (0%)	3 (14%)	0 (0%)	
4	1 (9.1%)	0 (0%)	13 (34%)		0 (0%)	0 (0%)	1 (20%)	11 (52%)	2 (17%)	
5	0 (0%)	0 (0%)	11 (29%)		0 (0%)	0 (0%)	0 (0%)	1 (4.8%)	10 (83%)	
Cognitive total raw score	55 (44, 68)	70 (49, 75)	32 (16, 43)	<0.001	73 (62, 75)	55 (48, 64)	43 (36, 44)	34 (30, 39)	12 (8, 16)	<0.001
Fine Motor Dependent	32 (22, 42)	42 (26, 48)	10 (2, 21)	<0.001	0.87 (0.74, 0.90)	0.63 (0.49, 0.71)	0.40 (0.29, 0.42)	0.23 (0.17, 0.33)	0.00 (0.00, 0.04)	<0.001
Fine Motor Independent	23 (22, 26)	27 (23, 29)	21 (15, 22)	0.002	0.69 (0.61, 0.76)	0.58 (0.56, 0.69)	0.56 (0.54, 0.56)	0.54 (0.51, 0.56)	0.31 (0.20, 0.36)	<0.001

^1^ Too many 0 s, so we cannot calculate a *p*-value here.

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
