# Peer review of "Manual Abilities and Cognition in Children with Cerebral Palsy: Do Fine Motor Skills Impact Cognition as Measured by the Bayley Scales of Infant Development?"

_behavsci, 2023, doi:10.3390/bs13070542_

Round 1

Reviewer 1 Report

The study “Manual Abilities and Cognition in Children with Cerebral Palsy: Do fine motor skills impact cognition?” is timely and valuable. In general, the manuscript is well written, however, minor changes are required. Please see my comments below.

Title

The current title is somewhat misleading: the link between fine motor performance and cognition could be different if the latter was measured by a different method. I think it would be clearer if the authors changed the title to something like: “Manual Abilities and Cognition in Children with Cerebral Palsy: Do fine motor skills impact cognition as measured by the Bayley Scales of Infant Development?”

Abstract

Remove statistical parameters from the Abstract.

p. 1, line 12: “Bayley Scale of Infant Development- 3rd edition (Bayley-III)” – change “Scale” to “Scales”, same for p. 4, line 112.

p. 1, lines 19-20: “The slope of FMD scores across MACS were significantly greater than the slope of FMI scores at baseline (p< 0.001).” I do not understand the meaning of this sentence. Please rephrase and clarify.

p 1, lines 21-22: “Children with lower manual ability scored lower in the cognitive domain.” – They scored lower at baseline, 9 months post-baseline, or at both time points?

Methods

p. 4, line 113: “was administered at baseline (N=61) and nine months post-baseline (n=28).” – N should ne changed for n, since both numbers represent a sample size. Same for Table 2.

Results

p. 5, line 183: “was observed after nine months post-baseline” – should be changed to either “was observed nine months post-baseline” or “was observed nine months after baseline”; in my opinion, the former sounds better.

p. 5, lines 185-186: “for both treatment groups (p-0,044 and 0.0012 for each treatment group respectively), but not significantly different between the two groups” – As far as I understand, there were no treatment groups in this study, just two types of Bayley questions. Please correct this issue here and thereafter.

p. 6, lines 190-19: “Rho fine motor dependent=-0.85, p<0.001 and Rho fine motor independent=-0.81” – should be changed to “Rho FMD=-0.85, p<0.001 and Rho FMI=-0.81”

p. 6, lines 191-192: “The slope of change in cognitive scores over MACS levels was significantly different (p<0.001).” – I do not understand this sentence, it sounds incomplete. Different between what and what?

p. 6, lines 194-195: “Similar patterns were observed at nine months post-baseline.” This sentence makes no sense to me. Similar patterns in the slope of change? But the slope shows the rate of change between the baseline and post-baseline. How can you talk about the slope at baseline and then the slope at post-baseline? Same for the slope description in the Abstract.

p. 7, lines 201-202:” A significant effect was found between the CP distribution and MACS level (p <0.001).” – Meaning what? And how was this analysis conducted (the description of this analysis is missing from the Methods)? Although CP distribution (groups) are presented in Table 1, they are not explicitly discussed in the text until the Results section.

p. 7, line 202: “was highly related” – should be changed to “significantly related”, p-values be themselves do not signal the magnitude of the effect.

p. 7, lines 203-205: “A similar association was found at nine months post-baseline, but not statistically significant and it may be due to the smaller sample size.” – This sentence makes no sense to me. How can you state that the association/relation is the same if in one case the relation is present and statistically significant, and in another one it is non-existent (not statistically significant)?

Author Response

We appreciate you and your precious time in reviewing our paper and providing valuable comments. It was your valuable and insightful comments that led to possible improvements in the current version. The authors have carefully considered the comments and tried our best to address every one of them. We hope the manuscript after careful revisions meet your high standards. The authors welcome further constructive comments if any.  Below we provide the point-by-point responses. All modifications in the manuscript have been highlighted in yellow.

Response to reviewer 1 comments 

Comment: The current title is somewhat misleading: the link between fine motor performance and cognition could be different if the latter was measured by a different method. I think it would be clearer if the authors changed the title to something like: “Manual Abilities and Cognition in Children with Cerebral Palsy: Do fine motor skills impact cognition as measured by the Bayley Scales of Infant Development?”

Response: The title has been modified to Manual Abilities and Cognition in Children with Cerebral Palsy: Do fine motor skills impact cognition as measured by the Bayley Scales of Infant Development?

Comment: Remove statistical parameters from the Abstract.

Response: p values and rho values were removed from the abstract.

Comment: p. 1, line 12: “Bayley Scale of Infant Development- 3rd edition (Bayley-III)” – change “Scale” to “Scales”, same for p. 4, line 112.

Response: Change is highlighted in yellow in the manuscript.

Comment: p. 1, lines 19-20: “The slope of FMD scores across MACS were significantly greater than the slope of FMI scores at baseline (p< 0.001).” I do not understand the meaning of this sentence. Please rephrase and clarify.

Response: Both FMD and FMI scores decreased as MACS level increased (worse function). There was a significant difference between the two slopes with the FMD scores having a steeper slope.

Comment: p 1, lines 21-22: “Children with lower manual ability scored lower in the cognitive domain.” – They scored lower at baseline, 9 months post-baseline, or at both time points?

Response: The new sentence is: Children with lower manual ability scored lower in the cognitive domain at baseline and 9 months post-baseline.

Comment: p. 4, line 113: “was administered at baseline (N=61) and nine months post-baseline (n=28).” – N should be changed for n, since both numbers represent a sample size. Same for Table 2.

Response: Change is highlighted in yellow in the manuscript.

Comment: p. 5, line 183: “was observed after nine months post-baseline” – should be changed to either “was observed nine months post-baseline” or “was observed nine months after baseline”; in my opinion, the former sounds better.

Response: The new sentence is: A similar relationship between cognitive score and MACS levels was observed nine months post-baseline (Rho=-0.80 and p values of <0.001).

Comment: p. 5, lines 185-186: “for both treatment groups (p-0,044 and 0.0012 for each treatment group respectively), but not significantly different between the two groups” – As far as I understand, there were no treatment groups in this study, just two types of Bayley questions. Please correct this issue here and thereafter.

Response:   The paragraph was removed from the result section.

Comment: p. 6, lines 190-19: “Rho fine motor dependent=-0.85, p<0.001 and Rho fine motor independent=-0.81” – should be changed to “Rho FMD=-0.85, p<0.001 and Rho FMI=-0.81”

 Response:. Change is highlighted in yellow in the manuscript.

Comment: p. 6, lines 191-192: “The slope of change in cognitive scores over MACS levels was significantly different (p<0.001).” – I do not understand this sentence, it sounds incomplete. Different between what and what?

Response:   The new sentence is: The decline between FMD and FMI cognitive scores over MACS levels was significantly different (p<0.001). Additionally, the decline of FMD scores across MACS levels is steeper than the FMI scores (0.09 vs 0.187 for FMD and FMI respectively, linear regression models, Figure 2).

Comment: p. 6, lines 194-195: “Similar patterns were observed at nine months post-baseline.” This sentence makes no sense to me. Similar patterns in the slope of change? But the slope shows the rate of change between the baseline and post-baseline. How can you talk about the slope at baseline and then the slope at post-baseline? Same for the slope description in the Abstract.

Response: The new sentence is: Similar patterns in the change slope in FMD and FMI cognitive scores over MACS levels were observed nine months post-baseline.

Comment: p. 7, lines 201-202:” A significant effect was found between the CP distribution and MACS level (p <0.001).” – Meaning what? And how was this analysis conducted (the description of this analysis is missing from the Methods)? Although CP distribution (groups) are presented in Table 1, they are not explicitly discussed in the text until the Results section.

Response: Change is highlighted in yellow.

  • p.5, line 170. Please read: In addition, using multiple regression analysis, we further investigated the impact of the distribution of CP and gross motor function level…
  • p.7, line 200. Please read: A significant association was found between the CP distribution and MACS level…
  • We clarified our secondary purposes regarding CP distribution “The secondary purpose was to investigate the impact of the distribution of CP and gross motor function levels (GMFCS) and factors such as gestational age, birth weight, NICU stay, age of CP diagnosis, and CP type on the relationship between manual ability levels and cognition.
  • Later, p.3 (Table 1) we present the sample’s descriptive data.
  • In the discussion section we mention that children with hemiplegia right or left score higher on cognitive domain than children with triplegia and quadriplegia. “Considering the CP distribution, our findings suggest that children with hemiplegia and diplegia CP, distributed mostly in MACS levels I to III, performed better on fine motor dependent items and, consequently, had better cognition performance than children with triplegia or quadriplegia, mostly assigned to levels IV and V. Children with triplegia or quadriplegia seem to perform better on fine motor independent items.”

Comment: p. 7, line 202: “was highly related” – should be changed to “significantly related”, p-values be themselves do not signal the magnitude of the effect.

Response: Change is highlighted in yellow. Please read: Moreover, the CP distribution was significantly related to the cognitive total raw score…

Comment: p. 7, lines 203-205: “A similar association was found at nine months post-baseline, but not statistically significant and it may be due to the smaller sample size.” – This sentence makes no sense to me. How can you state that the association/relation is the same if in one case the relation is present and statistically significant, and in another one it is non-existent (not statistically significant)?

Response: Change is highlighted in yellow. Please read: We did not find significant associations at nine months post-baseline.

Reviewer 2 Report

The present article deals with a topical issue and a social problem. The authors target pediatric clients after stroke. The article and its concept is very well prepared. The authors begin with an introductory chapter where they use a number of quality relevant sources. I would imagine that the authors will expand the introduction (Chapter 1) to include a subsection 1.1. where rehabilitation as such is mentioned. This is a rather general recommendation, as the introduction itself offers a wealth of information and introduces the reader very well to the chosen topic. The presentation of the results and their interpretation is also at a very erudite level. The authors evaluate the results of their own investigation and present interesting results. In the conclusion chapter, I recommend taking into account interdisciplinary articles that deal with similar issues (e.g. https://www.webofscience.com/wos/woscc/full-record/WOS:000794373400001 and others. I recommend adding multidisciplinary or holistic rehabilitation approaches here. This would make the conclusion chapter itself more versatile.

What is lacking here are the limitations of the study and a better formulation of conclusions and recommendations for practice. I therefore ask for additions.

Author Response

We appreciate you and your precious time in reviewing our paper and providing valuable comments. It was your valuable and insightful comments that led to possible improvements in the current version. The authors have carefully considered the comments and tried our best to address every one of them. We hope the manuscript after careful revisions meet your high standards. The authors welcome further constructive comments if any.  Below we provide the point-by-point responses. All modifications in the manuscript have been highlighted in yellow.

Response to reviewer 2 comments

Comment: The present article deals with a topical issue and a social problem. The authors target pediatric clients after stroke. The article and its concept is very well prepared. The authors begin with an introductory chapter where they use a number of quality relevant sources. I would imagine that the authors will expand the introduction (Chapter 1) to include a subsection 1.1. where rehabilitation as such is mentioned. This is a rather general recommendation, as the introduction itself offers a wealth of information and introduces the reader very well to the chosen topic. The presentation of the results and their interpretation is also at a very erudite level. The authors evaluate the results of their own investigation and present interesting results. In the conclusion chapter, I recommend taking into account interdisciplinary articles that deal with similar issues (e.g. https://www.webofscience.com/wos/woscc/full-record/WOS:000794373400001 and others. I recommend adding multidisciplinary or holistic rehabilitation approaches here. This would make the conclusion chapter itself more versatile.

What is lacking here are the limitations of the study and a better formulation of conclusions and recommendations for practice. I therefore ask for additions.

Response: The limitations of the study were presented on page 10, lines 315-324.  A few practice recommendations were mentioned on pages 9 and 10, in lines 283- 295, and 300-306. Thank you for suggestion articles and a multidisciplinary approach. We would have been interesting to explore more this aspect. However, we believe this approach was contemplated by suggesting an accurate, reliable quantification and a better understanding of real-life barriers children with CP face, especially between parents and rehabilitation professionals.

Reviewer 3 Report

Summary:

Based on cerebral palsy characteristics of cognition and motor impairments and musculoskeletal changes, this study explored the association between manual ability and cognition in children with CP. The results demonstrated that low levels of cognition are associated with high levels of motor impairment. The manuscript is interesting and well-written, however, I believe that some points are further clarified, as highlighted below.

Comments:

Abstract

The authors describe the results at the end of the abstract, but I recommend adding a brief conclusion.

Keywords

I suggest including ‘manual ability’ or ‘manual disability’ as keywords.

Introduction section

1. In the second paragraph, when the authors mention the Bayley scale for the first time, I suggest placing the acronym right after.

Methods section

1. The Bayley-III is used to assess a child's overall development. Why did the authors choose this scale? The WISC test, for example, is the gold standard for a robust assessment of cognition.

2. It is not clear how the Bayley-III can be used for children out of the age range.

Results section

1. Authors should better identify figures and tables in the results.

2. The authors highlighted some results that were not significant. Are there any uncommon conditions? Do you believe that only the difference in sample size contributed to this?

Discussion section

You make it clear that the sample size changed post-baseline. Does this not negatively affect the results of the study? Could the results be different with a larger sample?

Conclusions section

The authors should better highlight the study results in this section.

Author Response

We appreciate you and your precious time in reviewing our paper and providing valuable comments. It was your valuable and insightful comments that led to possible improvements in the current version. The authors have carefully considered the comments and tried our best to address every one of them. We hope the manuscript after careful revisions meet your high standards. The authors welcome further constructive comments if any.  Below we provide the point-by-point responses. All modifications in the manuscript have been highlighted in yellow.

Response to reviewer 3 comments

Comment: The authors describe the results at the end of the abstract, but I recommend adding a brief conclusion.

Response: The conclusion of the study was presented on page 1, lines 23-25.

Comment: I suggest including ‘manual ability’ or ‘manual disability’ as keywords.

Response: Change is highlighted in yellow in the manuscript.

Comment: In the second paragraph, when the authors mention the Bayley scale for the first time, I suggest placing the acronym right after.

Response: Change is highlighted in yellow. Please read: Many standardized cognitive assessments appropriate for children with cerebral palsy, such as the Bayley Scales of Infant Development- 3rd edition (Bayley-III) and…

Comment: The Bayley-III is used to assess a child's overall development. Why did the authors choose this scale? The WISC test, for example, is the gold standard for a robust assessment of cognition.

Response: This study is an exploratory analysis of data from a large clinical trial using Bayley-III.

Comment: It is not clear how the Bayley-III can be used for children out of the age range.

Response: According to the Bayley-III manual, the tool can be administered in children within the chronological age range or out of the age range but remaining in the developmental level for the Bayley-III. In this study, at 9 months post-baseline, besides those children in the chronological age range, 58.3% of the children were considered in the developmental level on the Bayley-III. Additionally, the cognitive age equivalent at this time-point for the children out of age range but in developmental level was (18.32 ± 13.65) in months and days. As recommended by the manual, the raw score was considered in the analysis.

Comment:. Authors should better identify figures and tables in the results.

Response: Changes are highlighted in yellow in the manuscript.

Comment: The authors highlighted some results that were not significant. Are there any uncommon conditions? Do you believe that only the difference in sample size contributed to this?

Response: As mentioned in the limitations of the study we believe as this study is an exploratory analysis of a set of data from a large clinical trial and other components of the trial may have confounded the results. Besides that, our sample had unequal sample sizes among MACS levels, possibly resulting in lower power for the subgroup analysis. In addition, although a comprehensive list of factors (birth-related, medical, and environmental) was analyzed, other demographics, such as parental education and maternal age, were not considered.

Comment: You make it clear that the sample size changed post-baseline. Does this not negatively affect the results of the study? Could the results be different with a larger sample?

Response: We believe the results could be different with a large sample size. This issue was addressed in the limitation paragraph.

Comment: The authors should better highlight the study results in this section.

Response:  Thank you for this suggestion. We have made changes to the results section to better explain the direction of associations and statistically significant relationships.